# Inhibition of PIKfyve Leads to Lysosomal Disorders via Dysregulation of mTOR Signaling

**DOI:** 10.3390/cells13110953

**Published:** 2024-05-30

**Authors:** Jianhong Xia, Haiyun Wang, Zhihang Zhong, Jun Jiang

**Affiliations:** 1CAS Key Laboratory of Regenerative Biology, Guangdong Provincial Key Laboratory of Stem Cell and Regenerative Medicine, Guangzhou Institutes of Biomedicine and Health, Chinese Academy of Sciences, Guangzhou 510530, China; jhxia2006@163.com (J.X.); sunny.haiyun@gmail.com (H.W.); 2Centre for Regenerative Medicine and Health, Hong Kong Institute of Science & Innovation, Chinese Academy of Sciences, Hong Kong SAR, China; 3State Key Laboratory of Swine and Poultry Breeding Industry, South China Agricultural University, Guangzhou 510642, China; zhongzhihanggood@163.com; 4Guangdong Provincial Key Laboratory of Protein Function and Regulation in Agricultural Organisms, College of Life Sciences, South China Agricultural University, Guangzhou 510642, China

**Keywords:** PIKfyve, mTOR, lysosome, macrophage, vacuolation

## Abstract

PIKfyve is an endosomal lipid kinase that synthesizes phosphatidylinositol 3,5-biphosphate from phosphatidylinositol 3-phsphate. Inhibition of PIKfyve activity leads to lysosomal enlargement and cytoplasmic vacuolation, attributed to impaired lysosomal fission processes and homeostasis. However, the precise molecular mechanisms underlying these effects remain a topic of debate. In this study, we present findings from PIKfyve-deficient zebrafish embryos, revealing enlarged macrophages with giant vacuoles reminiscent of lysosomal storage disorders. Treatment with mTOR inhibitors or effective knockout of mTOR partially reverses these abnormalities and extend the lifespan of mutant larvae. Further in vivo and in vitro mechanistic investigations provide evidence that PIKfyve activity is essential for mTOR shutdown during early zebrafish development and in cells cultured under serum-deprived conditions. These findings underscore the critical role of PIKfyve activity in regulating mTOR signaling and suggest potential therapeutic applications of PIKfyve inhibitors for the treatment of lysosomal storage disorders.

## 1. Introduction

Intracellular membranes are rich in various phosphoinositides (PtdInss) which play pivotal roles in determining organelle identity and regulating endolysosomal function, dynamics, and homeostasis [1]. The PtdIns repertoire on intracellular membranes is tightly regulated by specific sets of PtdIns kinases and phosphatases. Among these, PIKfyve stands out as an evolutionarily conserved endosomal PtdIns kinase extensively studied for its role in generating two rare PtdIns species: phosphatidylinositol-3,5-bisphosphate [PI(3,5)P2] and phosphatidylinositol-5-phosphate [PI(5)P] [2]. PIKfyve plays critical roles in the endolysosomal system, governing intracellular trafficking, membrane dynamics, and autophagy. Moreover, it orchestrates the subcellular distribution of key molecules such as epidermal growth factor receptor (EGFR) [3,4], tau aggregates [5], transcription factor EB (TFEB) [6,7], and tight junction protein claudins [8]. Notably, PIKfyve has garnered significant attention due to its pharmacological relevance in cancer [9,10], neurodegenerative diseases [5,11], lysosomal storage disorders [12], and susceptibility to various viral, bacterial and parasitic infections [13,14,15,16]. Recently, the PIKfyve inhibitor apilimod (Apm) has emerged as a potential therapeutic strategy against Ebola virus [13,15] and is currently undergoing clinical trials for the treatment of COVID-19 [1].

Accumulating evidence from patients and animal models pinpoints the critical functions of PIKfyve in development and diseases. Mutations in key sites within the phosphatidylinositol phosphate kinase (PIPKc) domain of the PIKfyve gene have been implicated in familial benign corneal dystrophy and congenital cataracts, although their precise pathogenesis remains elusive [17,18,19]. Global knockout of the PIKfyve gene in mice results in embryonic lethality due to impaired maternal nutrient uptake by the visceral endoderm [20,21]. Conditional PIKfyve deletion in mice using the Cre-loxP system has revealed its essential functions in multiple cell types, tissues, and organs, including the intestine [20], muscles [22], melanosome [23], platelets [24], and myeloid cells [25].

The most striking cellular phenotype observed upon pharmacological or genetic inhibition of PIKfyve is massive vacuolization, indicative of severe lysosomal stress and disruption of lysosomal balance and homeostasis. Despite recent advances, the precise molecular mechanisms underlying the accumulation of enlarged vacuoles in PIKfyve-deficient cells remain poorly understood.

The mechanistic target of rapamycin (mTOR) is a serine/threonine protein kinase that functions as a central regulator of cell metabolism, growth, proliferation and survival. Under nutrient-rich conditions, mTOR is recruited to the lysosomal surface, where it phosphorylates downstream targets such as S6K (S6 ribosomal protein kinase) and 4E-BP1 (eIF4E-binding protein 1), promoting protein synthesis and cell growth. Conversely, nutrient deprivation leads to mTOR inactivation, triggering cellular catabolism, reduced protein synthesis, and autophagy induction. Failure to finely tune mTOR activity results in metabolic dysregulation and diseases [26]. While there is substantial understanding of the molecular events leading to mTOR activation at the lysosome, little is known about what terminates mTOR signaling. Its dual role in activation and inactivation at the lysosome implicates mTOR in lysosomal homeostasis and prompts investigation into its reciprocal relationship with PIKfyve.

Previous studies on PIKfyve regulation of mTOR signaling have yielded conflicting results. Some suggest that PIKfyve is necessary for mTOR activation in adipocytes [27] and macrophages [25], while others report unaffected mTOR activity in PIKfyve-deficient cells under both normal and starvation conditions [6,7,28]. A recent study by Fitzian et al. highlighted the essential role of PIKfyve and its lipid product PI(3,5)P2 in the starvation-induced lysosomal recruitment of Tuberous sclerosis complex 2 (TSC2) and subsequent mTOR inactivation [29]. Further investigation is warranted to reconcile these apparent discrepancies in the literature and elucidate the complex and conflicting biological phenomena.

In this study, we explored the function of PIKfyve using zebrafish (*Danio rerio*) and cultured mammalian cells, uncovering evidence that PIKfyve is indispensable for mTOR inactivation both in vivo and in vitro. Additionally, we demonstrate that inhibition of mTOR activity confers protective effects against PIKfyve-deficiency-induced lysosomal defects.

## 2. Materials and Methods

### 2.1. Zebrafish Lines

Zebrafish were maintained as previously described [30]. Research on animals was performed with the approval of the animal ethics committee of Guangzhou Institutes of Biomedicine and Health, CAS. Wild-type (WT) TU, PIKfyve mutant (MU), and the following transgenic zebrafish lines were used in this study: Tg(*mpeg1*:EGFP), Tg(*mpx*:EGFP). Transgenic zebrafish lines were purchased from the Chinese Zebrafish Resource Center in Wuhan, China. Embryos were grown at 28.5 °C and kept under anesthesia with fish water containing 0.02% Tricaine (3-aminobenzoic acid ethyl ester, Sigma, St. Louis, MO, USA, E10521) during imaging; 0.003% PTU (1-phenyl-2-thiourea, Sigma, St. Louis, MO, USA, P7629) was added in fish water from 24 h post fertilization (hpf) to prevent pigment formation. The CRISPR/Cas9 system was used to generate PIKfyve and mTOR knockout zebrafish lines. The guide RNA(gRNA) targeting sequences were designed with the online tool CRISPRscan (https://www.crisprscan.org/ accessed on 4 March 2019) and are listed in Appendix A. Primers for genotyping are listed in Appendix A. The pCS2-nCas9n plasmid (Addgene Watertown, MA, USA, Plasmid #47929) was used to synthesize a zebrafish-codon-optimized Cas9 mRNA. To generate founders, 100 pg of gRNA and 300 pg of Cas9 mRNA were co-injected into one-cell-stage embryos. After incubation at 28.5 °C for 24 h, the genomic DNA was extracted from 30 injected embryos by heating the embryos at 94 °C for 20 min in lysis solution (50 mM NaOH). The reaction was terminated by adding 1/10 volume of 1 M Tris-HCl (pH 8.0). For PIKfyve mutant genotyping, DNA fragment was amplified from genomic DNA, and the PCR product was subjected to SfaNI digestion. F0 embryos with the highest editing efficiency were raised to sexual maturity, and heterozygous F1 fish were obtained and identified from the offspring of F0 fish using DNA sequencing. F1 fish carrying the same mutation were crossed to generate F2 homozygous mutant fish.

### 2.2. LysoTracker Red Staining

To stain the lysosomes in zebrafish, LysoTracker Red DND-99 dye (Invitrogen, Waltham, MA, USA, L7528) was diluted 1:25 in Phosphate-Buffered Saline (PBS) prior to injection of 2 nL into the hindbrain ventricle of larvae at 3 days post-fertilization (dpf). The embryos were rinsed with fresh fish water three times immediately before fluorescence microscopy imaging.

### 2.3. Cell Culture

RAW264.7 (ATCC, Manassas, VA, USA, TIB-71) and HeLa (ATCC, Manassas, VA, USA, CCL-2) cells were cultured in Dulbecco’s Modified Eagle Medium (DMEM) (high D-glucose; Invitrogen, Waltham, MA, USA) supplemented with 10% FBS (PAN-Biotech, Nbg. a. Inn, Bavaria, German) and 100 U/mL penicillin/streptomycin (Hyclone, Logan, UT, USA, SV30010) at 37 °C in a CO_2_ incubator (Thermo Fisher, Waltham, MA, USA). All of the cell lines were *Mycoplasma*-free as determined using a kit obtained from Lonza (Visp, Basel, Switzerland, LT07-318). Cell line identities were confirmed by short tandem repeat authentication.

### 2.4. Western Blot Analysis

Zebrafish embryos were collected and homogenized in RIPA buffer (20 mM Tris pH 8.0, 150 mM NaCl, 0.1% SDS, 0.5% Triton X-100) containing a complete protease inhibitor cocktail (Roche, Basel, Switzerland, 11697498001). Protein samples were loaded onto 12% SDS–PAGE mini-gels before transfer onto PVDF membranes (Millipore, Burlington, MA, USA, IPVH00010). Membranes were blocked with 5% milk in TBS, 0.1% Tween-20 for 1 h at room temperature (RT), then incubated with primary antibodies overnight at 4 °C. Bound antibodies were detected with horseradish peroxidase (HRP)-coupled secondary antibodies (incubated for 1 h at RT) and an ECL system (Millipore, Burlington, MA, USA, WBKLS0500). Antibodies used in this study are listed in Appendix A.

### 2.5. Drug Treatment

Drug treatment was performed in 12-well plates. Drugs were administered via the fish water or culture medium for zebrafish embryos or cultured cells, respectively. Chemical compounds used in this study are listed in Appendix A and were all procured from Selleck Chemicals (Houston, TX, USA).

### 2.6. Immunofluorescence Staining

Cells grown on glass coverslips were fixed in 4% paraformaldehyde/PBS for 30 min and permeabilized in 0.2% Triton X-100/PBS for 15 min at RT. After two brief washes in PBS, cells were blocked in blocking buffer (5% FBS/0.2% Triton X-100/PBS) for 30 min, then incubated with primary antibodies diluted in blocking buffer overnight at 4 °C. After four washes with PBS, cells were incubated with secondary antibodies (Alexa Fluor 488/568-conjugated goat anti-mouse/rabbit IgG, 1:400) (Molecular Probes, Thermo Fisher, Waltham, MA, USA) in blocking buffer for 1 h at RT. Cells were then washed four times with PBS, counterstained with DAPI (10 µg/mL) (Invitrogen, Waltham, MA, USA, D3571) for 2 min, and mounted onto glass slides with anti-fade solution (Life Technologies, Carlsbad, CA, USA, P36934) and visualized using a Zeiss 710 NLO confocal microscope (Carl Zeiss, Oberkochen, Germany). Co-localization analysis was performed by ImageJ plugin Coloc2 (version 1.53).

For whole-mount immuno-fluorescence staining on zebrafish embryos, larvae were fixed in 4% paraformaldehyde at 4 °C overnight. Fixed larvae were dehydrated and stored in 100% methanol at −20 °C. After rehydration and several washes with PBST (1X PBS with 0.1% Tween-20), embryos were then incubated in cold acetone for 7 min at −20 °C, washed with PBST, and blocked in blocking buffer for 1 h at RT. Embryos were incubated with indicated primary antibodies diluted in blocking buffer overnight at 4 °C, washed with PBST 4 times, and incubated with secondary antibodies (Alexa Fluor 488/568-conjugated goat anti-rabbit IgG, Molecular Probes, 1:1000) for 2 h at RT. After several washes with PBST, embryos were counterstained with DAPI (10 µg/mL) for 2 min. Embryos were flat-mounted and imaged using a Zeiss 710 NLO confocal microscope. Fluorescence images were manipulated with Adobe Photoshop CC 2019 (Adobe Inc., San Jose, CA, USA).

### 2.7. Real-Time Quantitative PCR Analysis

Embryos from incrossing between heterozygous fish were collected at 3 dpf. About 1/3 of the tail from individuals were clipped and genotyped, whereas the rest of the body was placed into individual PCR tubes with Trizol Reagent (Invitrogen Carlsbad, CA, USA, 15596018) and homogenized. The lysates were placed at −80 °C until genotyping was completed. Genotyped embryos were grouped into WT siblings and heterozygous and homozygous mutants, yielding 3 groups from 3 different clutches (3 biological replicates). Total RNA was isolated from ~30 zebrafish embryos with 1 mL TRIzol Reagent. Total RNA was reverse transcribed using the ReverTra Ace transcriptase (TOYOBO, Osaka, Japan, TRT-101) and oligo dT primers. After complementary DNA (cDNA) synthesis, real-time quantitative PCR (RT-qPCR) was performed with SYBR Green Supermix (BIO-RAD, Hercules, CA, USA, 172-5274) in a CFX96 real-time system (BIO-RAD). Plots show the results of three biological replicates (3 pools of embryos per developmental stage or treatment, and each pool consisted of 30 embryos). Each biological replicate was run in triplicate, and average values were plotted. Data were normalized to β-actin for ΔΔCt analysis. Primer sequences used for RT-qPCR are listed in Appendix A.

### 2.8. Microscopy and Fluorescent Pixel Count

Embryos were anaesthetized in 0.02% Tricaine, embedded in 1% low-melting point agarose (Sigma, St. Louis, MO, USA, A9414) in glass-bottomed dishes, and covered with 1 mL of fish water containing Tricaine. Embryos were imaged using a microscope (Zeiss Axio A1, Oberkochen, Germany) with an AxioCam MRc camera or a Zeiss 710 NLO confocal microscope. Maximal intensity projections of confocal micrograph z-stacks were shown. Time-lapse images were taken at 3 min intervals for 40 cycles. Data were acquired with ZEN z-stack module and time-lapse module (Zeiss). Macrophage tracks were generated and the average speed was measured using Imaris 9.0 (Bitplane Scientific Software). Quantification of total fluorescent pixels in the images of individual embryos was performed using ImageJ software (version 1.53) as described [31].

### 2.9. Neutral Red Staining

Neutral Red (NR) staining was performed by incubating embryos in 2.5 μg/mL NR (Sangon Biotech, Shanghai, China, A600652) solution at 28.5 °C in the dark for 6–8 h, followed by two water changes, and then observed and imaged using a stereo microscope.

### 2.10. Statistical Analysis

All statistical analyses were performed using GraphPad Prism software (Version 8.4; GraphPad, San Diego, CA, USA). Differences between two groups were analyzed using unpaired, two-tailed Student’s *t*-test. Differences between more than two groups were analyzed using one-way or two-way ANOVA. Differences between survival curves were analyzed using log-rank test. Significance (*p*-value) is indicated as the following: ns, no significant difference; * *p* < 0.05; ** *p* < 0.01; *** *p* < 0.001, **** *p* < 0.0001. Error bars: mean ± standard deviation.

## 3. Results

### 3.1. Generation of PIKfyve Mutant Zebrafish Using CRISPR/Cas9

The zebrafish PIKfyve gene, identified approximately a decade ago, exhibits widespread expression during early development [18]. Utilizing CRISPR/Cas9 technology, we generated PIKfyve mutant zebrafish lines. A gRNA was designed to target exon 38 of zebrafish PIKfyve gene (GenBank Accession No. NM_001127305), encoding the PIPKc lipid kinase domain (Appendix A) crucial for catalyzing the Phosphatidylinositol 3-phosphate [PI(3)P] phosphorylation to produce PI(3,5)P_2_. Two independent mutation alleles (MU1 and MU2) were maintained, introducing frame-shifted transcripts with premature stop codons. The MU1 allele had a 5 bp deletion, while MU2 had an 11 bp insertion (+13-2). Sanger sequencing confirmed these mutations (Appendix A). Since preliminary analyses showed that the two mutant lines displayed similar and consistent developmental defects, we used MU1 for the subsequent experiments hereafter.

### 3.2. General Morphological Defects in PIKfyve Mutant Zebrafish

Heterozygous fish were cross-mated to produce F2 homozygous mutants and their phenotypes were investigated. Homozygous PIKfyve mutants and WT siblings were morphologically indistinguishable before 3 dpf. At 3–4 dpf, multiple characteristic morphological abnormalities were observed in PIKfyve mutants, including: (1) giant vacuole-like structures that appeared in the caudal hematopoietic tissue (CHT) (Figure 1A, arrow) and (2) massive cell death/degeneration in the brain (Figure 1A, asterisk), pharynx, esophagus, and intestinal tract. All mutant embryos succumbed to these abnormalities by 7 dpf. Notably, *PIKfyve* +/− heterozygous zebrafish were viable, developed into fertile adults, and exhibited no discernible phenotypes.

To assess the impact of PIKfyve mutations at the molecular level, we quantified the abundance of PIKfyve mRNA using RT-qPCR. In the homozygous mutants, we observed a significant reduction in PIKfyve transcripts to less than 40% of those measured in WT siblings (Figure 1B). The presence of premature stop codons in the edited PIKfyve mRNAs may lead to the production of truncated proteins or the degradation of mRNAs through nonsense-mediated mRNA decay (NMD) [32]. NMD is an evolutionarily conserved mRNA degradation pathway that eliminates mRNAs with a premature termination codon (PTC) arising from mutations, alternative splicing, or other events in cells [33,34]. High throughput systematic analysis of half-lives of the aberrant mRNAs containing PTCs demonstrate that a small percent escape surveillance and do not degrade [35]. In this study, the remaining ~40% of PIKfyve transcripts with PTCs may survive due to evasion of the NMD mechanism, whereas they were unable to translate into a functional protein.

### 3.3. PIKfyve Mutation Leads to Lysosomal Disorders in Macrophages

Our investigation aimed to identify the cell types harboring giant vacuole-like structures within the CHT. The zebrafish CHT comprises various cell types, including macrophages and neutrophils [36], which are major components of the innate immune system. To pinpoint the lineage of cells containing these vacuoles, we utilized Tg(*mpeg1*:EGFP) and Tg(*mpx*:EGFP) transgenic zebrafish lines on the PIKfyve mutant background, where macrophages and neutrophils were labeled with EGFP, respectively.

There was no significant difference in the number of macrophages or neutrophils in mutants compared with WT siblings at 3 dpf (Figure 1C–F). However, most of the macrophages in mutants were enlarged and discoid in shape (Figure 1C,D). In contrast to the enlarged macrophages, the mutant neutrophils (Figure 1E,F) displayed roughly normal morphology at this developmental stage. The differential response of macrophages and neutrophils to PIKfyve deficiency suggests that PIKfyve and its lipid products may have different effects on distinct immune cell populations, potentially reflecting differences in their lysosomal physiology, metabolic demands, or signaling pathways.

Microglia are a specialized population of resident macrophages in the central nervous system. NR staining demonstrated that a proportion of mutant microglia were larger in size and stained darker (Figure 1G), while the number of microglia did not change significantly in mutant larvae (Figure 1H).

Further analysis revealed that the enlarged macrophages in mutant zebrafish contained giant vacuoles, identified as fused lysosomes by staining with LysoTracker (Figure 2A,B) or anti-lysosome-associated membrane protein 1 (LAMP1) antibody (Figure 2C,D). Additionally, the homeostatic migration speed of mutant macrophages was significantly reduced (Figure 2E,F and Appendix A). These aberrant morphological and behavioral characteristics of mutant macrophages resemble those observed in lysosomal storage disorders in zebrafish [37] and mouse [38] models, as well as in human patients [39].

Overall, these findings underscore the crucial role of PIKfyve in lysosomal biogenesis and homeostasis in zebrafish macrophages.

### 3.4. The Defects in PIKfyve Mutants Are Dependent on Sustained mTOR Activity

In our pursuit of potential treatments for PIKfyve mutants, we conducted a thorough screening of various chemical compounds, including apoptosis or necrosis inhibitors, mTOR inhibitors, and autophagy modulators, among others (Appendix A). Remarkably, our investigation revealed that treatment with two mTOR inhibitors, rapamycin and Torin-1, significantly alleviated the vacuolation defects observed in PIKfyve mutants. Administering these mTOR inhibitors to zebrafish larvae from 2 dpf, before symptom onset, effectively ameliorated both the overall and lysosomal phenotypes in PIKfyve mutants by 5 dpf (Figure 3A–C). Notably, the lifespan of mutant larvae was extended to approximately 9 dpf with rapamycin or Torin-1 treatment, in sharp contrast to the control group treated with Dimethyl Sulfoxide (DMSO) vehicle, where the survival period was approximately 7 dpf (Figure 3B). In addition, macrophage diameter in WT siblings was not affected by pharmacological inhibition of mTOR (Appendix A).

In addition to chemical approaches using small-molecule inhibitors, we utilized genetic approaches to knockout mTOR using CRISPR/Cas9 gene editing technology. Microhomology-mediated end joining (MMEJ)-based gRNAs enable highly effective CRISPR/Cas9 mutagenesis and rapid generation of zebrafish F0 knockouts [40]. According to a previous report [41], a highly efficient MMEJ-based gRNA for mTOR gene was designed. An optimized mixture of Cas9 protein/gRNA was injected into zebrafish embryos at the one-cell stage; larvae injected with Cas9 protein alone (with no gRNA) were used as control. The resultant KO score was evaluated by the SYNTHEGO website (https://ice.synthego.com/ accessed on 12 April 2021) (Appendix A). Raw sequence alignment of the whole PCR amplicon showed that about 50% of reads were a 10 bp deletion allele. The mutagenesis rate in the F0 zebrafish is about 80%, indicating that most cells contained biallelic mutations. As expected, effective F0 knockout of mTOR gene produced a similar rescue effect to mTOR inhibitor treatment. The overall and vacuolation phenotypes were mitigated and the longevity was significantly increased in the PIKfyve homozygous mutants with effective mTOR knockout (Figure 3A–C).

We reasoned that lysosomal enlargement in the PIKfyve mutant cells would affect the proper activation/inactivation of mTOR signaling. To determine whether PIKfyve mutation causes abnormalities of mTOR signaling during the development of zebrafish, we evaluated the phosphorylation status of S6K and 4E-BP1, two well-defined mTOR downstream effectors, using whole embryo lysates at different developmental stages. The phosphorylation of S6K and 4E-BP1 decreased gradually from 3 dpf to 5 dpf in WT siblings, indicating the shutdown of mTOR signaling in response to the changing nutrient availability during yolk consumption. In contrast, high levels of phosphorylated S6K and 4E-BP1 were sustained in PIKfyve mutant embryos from 3 dpf to 5 dpf (Figure 4A,B). Additionally, phospho-S6K staining was markedly elevated in the enlarged macrophages in the PIKfyve mutants at 5 dpf (Figure 4E). Rapamycin treatment dramatically reduced the levels of phospho-S6K as revealed by Western blot (Figure 4C,D) and immuno-fluorescence staining (Figure 4E). These results illustrated the effectiveness of rapamycin treatment in shutting down the mTOR pathway and the specificity of the staining signals in zebrafish samples.

Aside from the morphological alteration, we also assessed the changes at the molecular level in PIKfyve mutant larvae and PIKfyve-deficient cells. We analyzed LAMP1 levels in total lysates of WT and PIKfyve mutant larvae at 5 dpf. LAMP1 protein levels were significantly increased in PIKfyve mutants relative to WT siblings and returned to baseline with rapamycin treatment (Figure 5A,B). Similar results were obtained in murine macrophage cells (RAW264.7) treated with Apm, a potent and specific PIKfyve inhibitor (Figure 5C,D). Staining of lysosomal compartments with LysoTracker or anti-LAMP1 antibody revealed significantly enlarged lysosomal area in PIKfyve-deficient cells. Rapamycin treatment recovered these changes (Figure 5E).

In summary, the accumulation of fused giant lysosomes caused by PIKfyve deficiency is attenuated by mTOR inhibition, both in zebrafish embryos and in cultured cells.

Pharmacological or genetic mTOR inhibition efficiently improve the phenotypic and molecular consequences in PIKfyve-deficient cells.

### 3.5. PIKfyve Is Required for Starvation-Induced Shutdown of mTOR in Cultured Cells

In contrast to the well-studied mTOR activation machinery on the lysosomal surface, much less is known about the mechanisms of mTOR shutdown. We found that PIKfyve activity is indispensable for the attenuation of mTOR signaling during zebrafish embryonic development. To elucidate the molecular mechanisms underlying this observation, we took advantage of the mammalian cell culture system and investigated whether the inactivation of mTOR signaling under serum starvation is dependent on PIKfyve activity.

We assessed temporal regulation of mTOR signaling under conditions of serum starvation in presence or absence of PIKfyve inhibitor. Apm strongly delayed the attenuation of the mTOR signaling pathway under serum starvation conditions, as evaluated by the phosphorylation of 4E-BP1 and S6K (Figure 6A,B).

TSC2 is a critical negative regulator of mTOR signaling. TSC2 translocates to lysosome to inactivate mTOR in response to a variety of cellular stresses including starvation, osmostress, and energetic stress [42]. Previous works suggest that TSC2 lysosomal translocation and subsequent inactivation of mTOR upon starvation depend on PIKfyve and its lipid product PI(3,5)P_2_ [29]. To achieve a better understanding of the molecular mechanisms of PIKfyve regulation of mTOR signaling, we investigated the lysosomal recruitment of TSC2 during starvation-induced mTOR inactivation. In line with previous reports [29,42], in control cells treated with DMSO vehicle, starvation led to the relocalization of TSC2 to the LAMP1-positive lysosomal compartments. In contrast, TSC2 remained diffusely distributed in the cytoplasm in Apm-treated cells, even upon starvation (Figure 6C,D).

In conclusion, our findings support a model in which PIKfyve regulates mTOR signaling by translocating TSC2 to the lysosomes and inactivating mTOR (Figure 6E). mTOR activity is necessary for PIKfyve-related lysosomal defects and mTOR inhibition reduces accumulation of fused giant lysosomes in PIKfyve-deficient cells, improving their phenotypic and molecular consequences.

## 4. Discussion

To the best of our knowledge, our study represents the first in vivo demonstration of PIKfyve’s pivotal role in regulating mTOR shutdown. We observe sustained mTOR levels in PIKfyve mutant zebrafish. Furthermore, inhibition of mTOR yields an amelioration of the vacuolation phenotype in the PIKfyve-deficient models in vivo and in vitro. Further mechanistic investigations unveiled that PIKfyve activity is crucial for mTOR shutdown under starvation stress in cell culture systems.

Despite nearly 100% phenotypic penetration of homozygous MU zebrafish, we still lack direct evidence of complete knockout of the target gene at the molecular level. To measure the changes of PIKfyve or mTOR protein in the mutants, Western blot was applied in lysates from WT, PIKfyve MU, and mTOR-KO zebrafish embryos. Compared to mammalian models, zebrafish research currently suffers from a lack of specific antibodies. We have tested some commercially available antibodies against human PIKfyve (Invitrogen PA5-67981, Abnova H00200576-M01, SantaCruz sc-100408) or mTOR (CST, #2983), but none of them exhibit cross-reactivity with zebrafish samples. So, we could not experimentally verify the changes of target protein in PIKfyve or mTOR mutants. Furthermore, a direct readout of PIKfyve activity will help to validate the functional impact of PIKfyve mutation. We plan to perform in vitro kinase assays by measuring PI(3,5)P2 levels using zebrafish lysates [43]. In addition, referring to the previously published PI(3)P reporter zebrafish line in our lab [30], we designed a fluorescent reporter system for real-time monitoring of PI(3,5)P2 in zebrafish embryos. A transgenic zebrafish line expressing the PI(3,5)P2 biosensor (GFP-2xMLN1) [44] driven by the ubiquitin promoter is being constructed. These assays will provide a direct readout of PIKfyve enzymatic activity in the WT and MU zebrafish larvae, allowing us to validate the functional impact of PIKfyve knockout.

While our in vitro cell culture systems provided valuable insights, we lacked data on in vivo TSC2 localization differences in WT and MU zebrafish due to the absence of effective specific antibodies. Consequently, our next steps involve generating novel transgenic zebrafish reporter lines in which TSC2 is fluorescently labeled with EGFP, facilitating real-time in vivo monitoring of TSC2 localization changes following PIKfyve inhibition. It has been shown that VPS34 forms a protein complex with PIKfyve and TSC1, which disrupts the TSC1/TSC2 complex, resulting in ubiquitination and degradation of TSC2 and consequent activation of the Rheb-mTORC1 axis [45]. In our model presented in this study, inhibition of PIKfyve prevents the lysosomal recruitment of TSC2, thereby impeding mTOR inactivation. This corroborating information leads us a hypothesis that inhibition of PIKfyve could lead to activation of mTOR via disruption of the TSC1/TSC2 complex. Lysosomes play pivotal roles in both mTOR signaling activation and shutdown. While PIKfyve’s role in lysosomal homeostasis is well established, its regulation of mTOR activity on the lysosome surface has been subject to debate. Conflicting data may stem from different animal models and experimental approaches. For instance, Hasegawa et al. [6] reported that PIKfyve does not regulate the activity of mTOR under starvation conditions. Their data were primarily taken at two static time points: before and after 2 h of starvation. In contrast, Fitzian et al. [29] performed similar experiments but with deeper and more nuanced time course analysis during this 2 h time window, and they concluded that PIKfyve is indispensable for starvation-induced mTOR inactivation, which was further supported by our present study. These observations suggest that PIKfyve regulation of mTOR activity is a complex and dynamic process rather than a static event.

Aberrant activation of the mTOR signaling pathway has emerged as a significant driver of tumor growth and progression, prompting considerable interest in mTOR as a target for anti-tumor drug development [26]. This has led to the development of numerous mTOR inhibitors with varying mechanisms of action, some of which are currently undergoing clinical trials for different types of human cancers. While zebrafish embryos offer a valuable platform for high-throughput drug screening [46], WT embryos are not optimal for screening mTOR inhibitors due to the lack of observable defects before 6 dpf [47] and inefficiencies in drug absorption in older larvae. In our study, we reveal PIKfyve as a key negative regulator of mTOR signaling during zebrafish development. Through a combination of in vivo and in vitro genetic and pharmacological analyses, we demonstrate that sustained activation of mTOR activity is a major contributor to the lysosomal defects observed in PIKfyve mutants. Importantly, we find that inhibiting mTOR activity effectively rescues the phenotype of PIKfyve mutant zebrafish embryos, a rescue effect easily discernible under a stereomicroscope. Therefore, in conjunction with traditional cell-based screening assays, PIKfyve mutant zebrafish embryos provide a rapid and efficient in vivo screening platform for identifying novel mTOR inhibitors.

PIKfyve plays a crucial role in regulating intracellular membrane trafficking and organelle homeostasis. In recent years, accumulating evidence has highlighted the significance of PIKfyve in autophagy regulation, a highly conserved cellular process involved in the degradation and recycling of cellular components. Inhibition of PIKfyve induces lysosome-associated cytoplasmic vacuolation in hepatocellular carcinoma cells [48]. Also, glucose starvation can induce autophagy by ULK1-mediated activation of PIKfyve [49], and trehalose can activate PIKfyve, leading to TFEB nuclear translocation and autophagy induction [50]. PIKfyve has been implicated in the regulation of TFEB, a master regulator of lysosomal biogenesis and autophagy. Studies have shown that PIKfyve inhibition leads to mTORC1-dependent phosphorylation and cytoplasmic retention of TFEB, thereby inhibiting its transcriptional activity and impairing lysosome biogenesis. Conversely, activation of PIKfyve promotes TFEB nuclear translocation and enhances autophagic activity. These findings underscore the multifaceted role of PIKfyve in coordinating cellular responses to nutrient availability and stress, with implications for autophagy induction and lysosome function. Given the emerging role of autophagy dysregulation in various diseases, including cancer and neurodegenerative disorders, understanding the interplay between PIKfyve and autophagy pathways may offer novel opportunities for therapeutic intervention.

Moreover, recent studies have linked PIKfyve mutations to congenital cataracts in both humans and zebrafish. Treatment with the V-ATPase inhibitor Bafilomycin A1 (Baf-A1) has shown promise in alleviating the aberrant vacuolation phenotype in the lens of mutant zebrafish [19]. Building upon these findings, our study identifies therapeutic effects of mTOR inhibition on PIKfyve mutants. These discoveries offer new avenues for the development of novel treatment strategies for PIKfyve-dependent genetic diseases, including congenital cataracts and lysosomal storage disorders.

## Figures and Tables

**Figure 1 cells-13-00953-f001:**
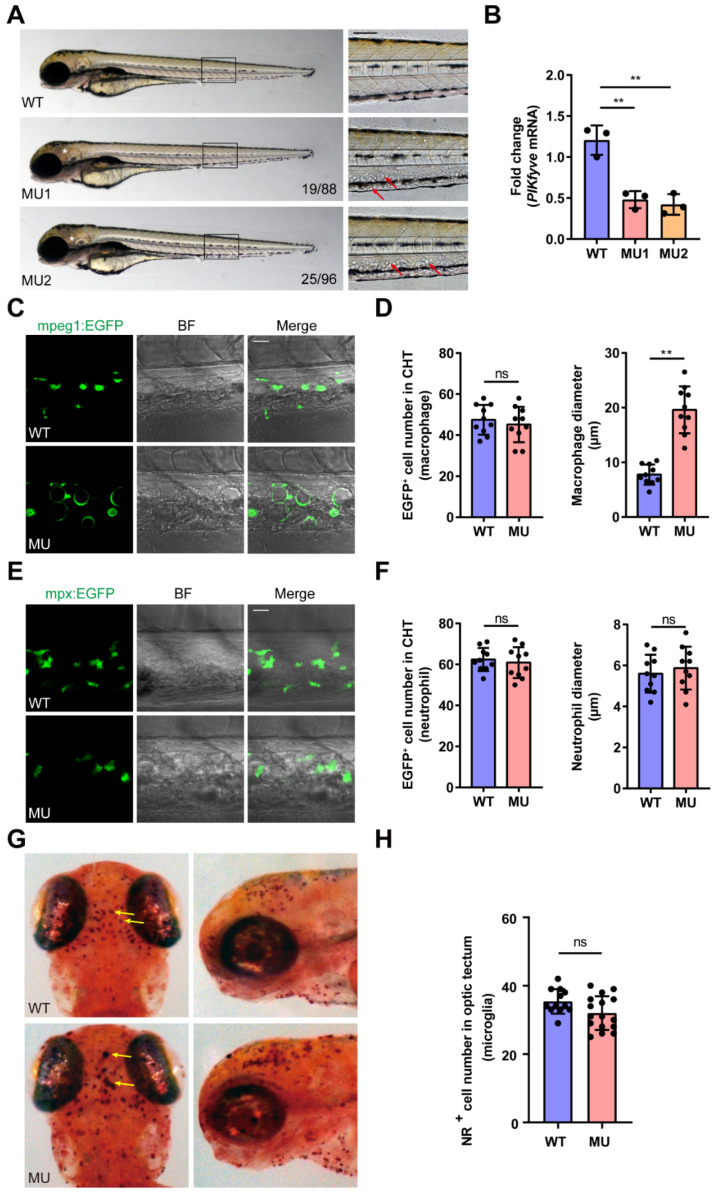
Defects in macrophages in PIKfyve mutant zebrafish embryos. (**A**) Representative images of WT and MU embryos at 3 dpf. The arrows show the cells with giant vacuole-like structures in the CHT. Cellular death in the mutant brains was evidenced by grey tissue (asterisk). (**B**) The relative expression levels of PIKfyve in the WT zebrafish and the homozygous mutants (MU1, MU2) were quantified by RT-qPCR. The data were analyzed by 2^−∆∆CT^ method and *ꞵ-actin* was used as an internal control (** *p* < 0.01). (**C**–**F**) PIKfyve mutation was introduced into transgenic zebrfish lines in which macrophages (*mpeg1*:EGFP) (**C**,**D**) or neutrophils (*mpx*:EGFP) (**E**,**F**) were respectively highlighted with EGFP. In the mutant embryos, the macrophages were enlarged with giant vacuoles, while the neutrophils appeared normal. Scale bar, 20 µm. (**D**) Quantification of macrophage cell number (**left panel**) and macrophage diameter (**right panel**) in the CHT (ns, non-significant; ** *p* < 0.01). (**F**) Quantification of neutrophil cell number (**left panel**) and neutrophil diameter (**right panel**) in the CHT (ns, non-significant). (**G**,**H**) NR staining of microglia in WT siblings and PIKfyve mutants at 3 dpf. (**G**) Dorsal and lateral view of the microglia in the optic tectum of WT and MU larvae at 3 dpf stained with NR. The yellow arrows show the microglia stained with NR. (**H**) Quantification of NR-positive microglia cell number in the optic tectum of WT and MU larvae at 3 dpf (ns, non-significant).

**Figure 2 cells-13-00953-f002:**
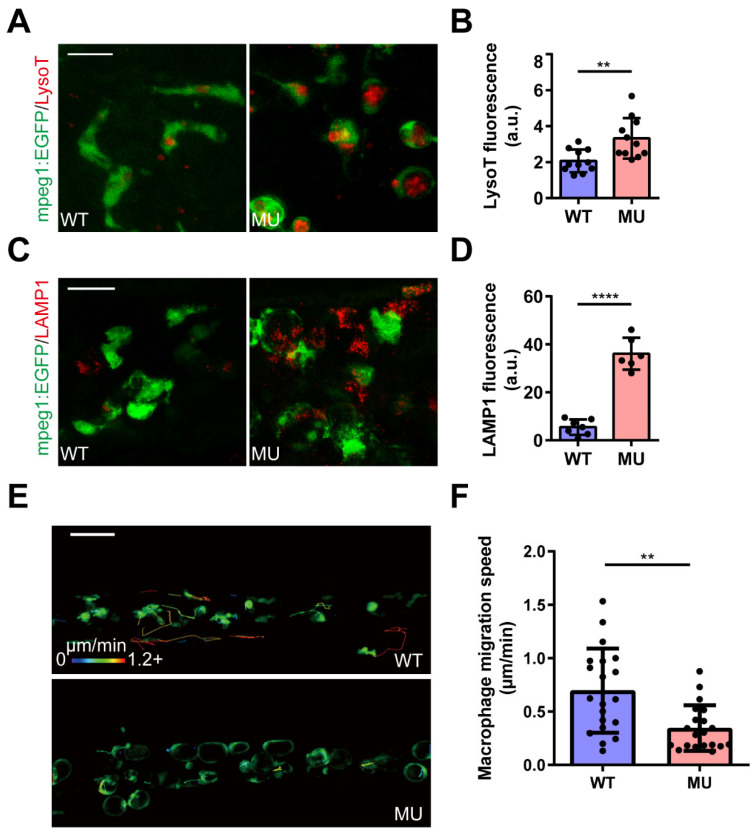
PIKfyve deficiency leads to the accumulation of enlarged LAMP1-positive compartments in macrophages. (**A**–**D**) The giant vacuoles in mutant macrophages were revealed to be fused lysosomes by staining with LysoTracker (**A**) or anti-LAMP1 antibody (**C**). (**B**) and (**D**) respectively show the quantification of LysoTracker and LAMP1 fluorescence intensity (** *p* < 0.01, **** *p* < 0.0001). Scale bar, 20 µm. (**E**) Representative confocal images and tracks of macrophage movement in the CHT of WT and MU zebrafish larvae at 3 dpf. Tracks are coded for speed. Scale bar, 50 µm. (**F**) Quantification of migration speed of macrophages in (**E**) (** *p* < 0.01).

**Figure 3 cells-13-00953-f003:**
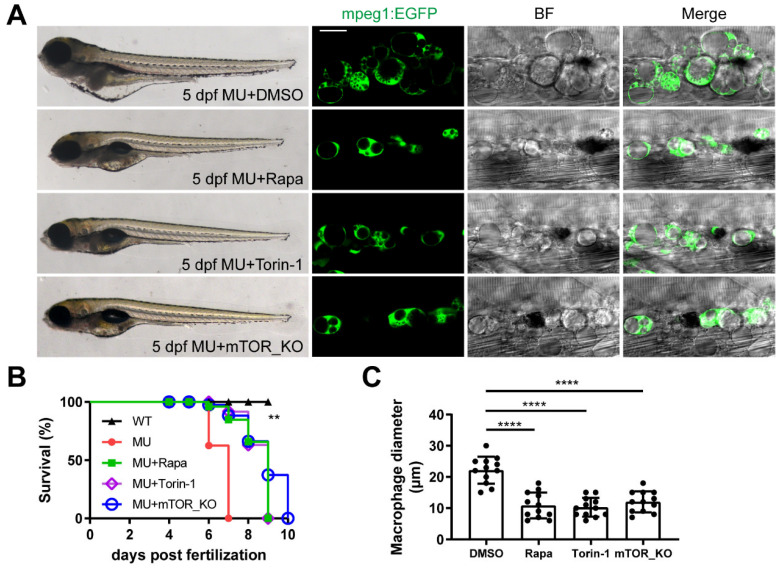
The defects in PIKfyve mutants are dependent on sustained mTOR activation. (**A**) mTOR inhibitor (Rapa or Torin-1) treatment or effective mTOR knockout (mTOR_KO) attenuates the developmental defects in PIKfyve mutants. Scale bar, 20 µm. (**B**) Survival curve showing that the lifespan of MU embryos can be extended when mTOR is inhibited by chemical (Rapa, Torin-1) or genetic (mTOR_KO) approaches (N = 3, *n* = 30 per group). (** *p* < 0.01, log-rank test). (**C**) Quantification of diameter of macrophages in MU zebrafish embryos treated as in (**A**) (**** *p* < 0.0001, one-way ANOVA).

**Figure 4 cells-13-00953-f004:**
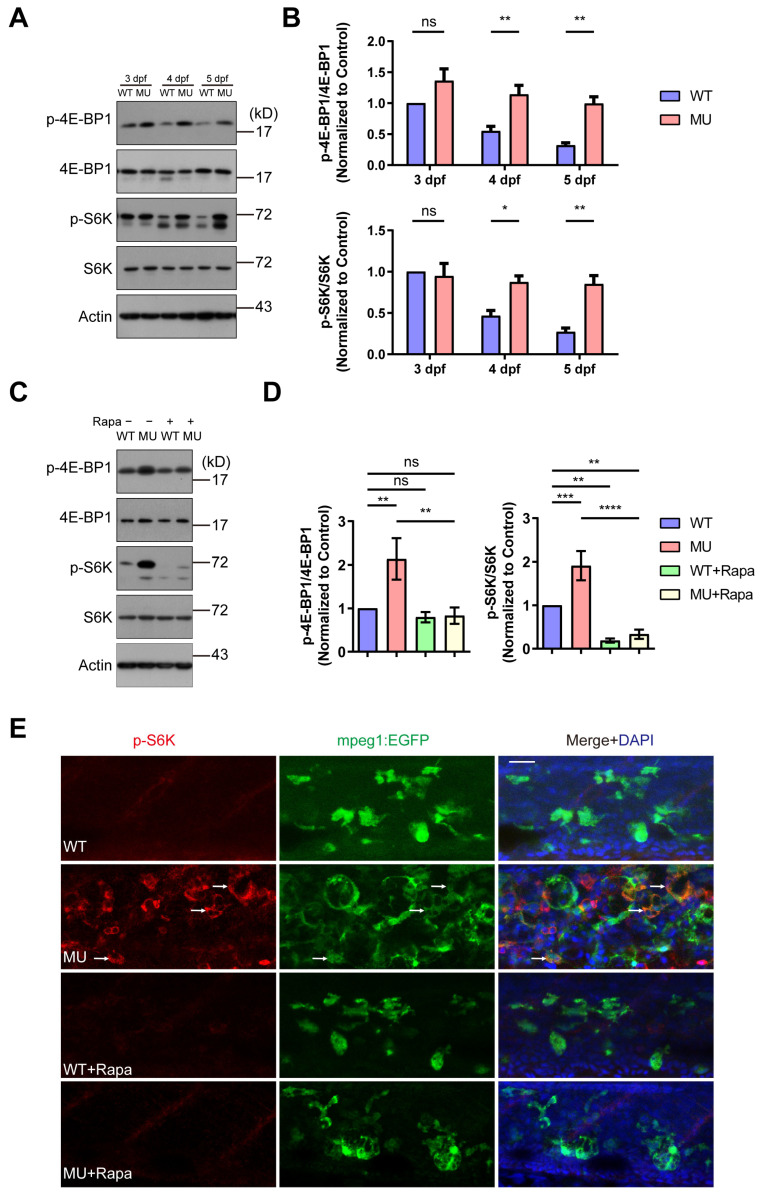
Molecular evidence for the hyperactivation of mTOR signaling in PIKfyve mutant zebrafish. (**A**) Western blot analysis showing the sustained mTOR activity during early development in PIKfyve mutant zebrafish embryos. (**B**) Densitometric analysis of phosphorylation status of 4E-BP1 and S6K in (**A**) (ns, non-significant; * *p* < 0.05, ** *p* < 0.01; two-way ANOVA). (**C**) Western blot analysis of mTOR activity in WT and MU zebrafish embryos at 5 dpf with or without rapamycin treatment. (**D**) Densitometric analysis of phosphorylation status of 4E-BP1 and S6K in (**C**) (ns, non-significant; ** *p* < 0.01, *** *p* < 0.001, **** *p* < 0.0001; two-way ANOVA). (**E**) WT and MU zebrafish embryos with or without rapamycin treatment were fixed at 5 dpf, permeabilized, and immuno-stained with phospho-S6K antibody. Arrows denote the macrophages with intense phospho-S6K staining. Scale bar, 20 µm.

**Figure 5 cells-13-00953-f005:**
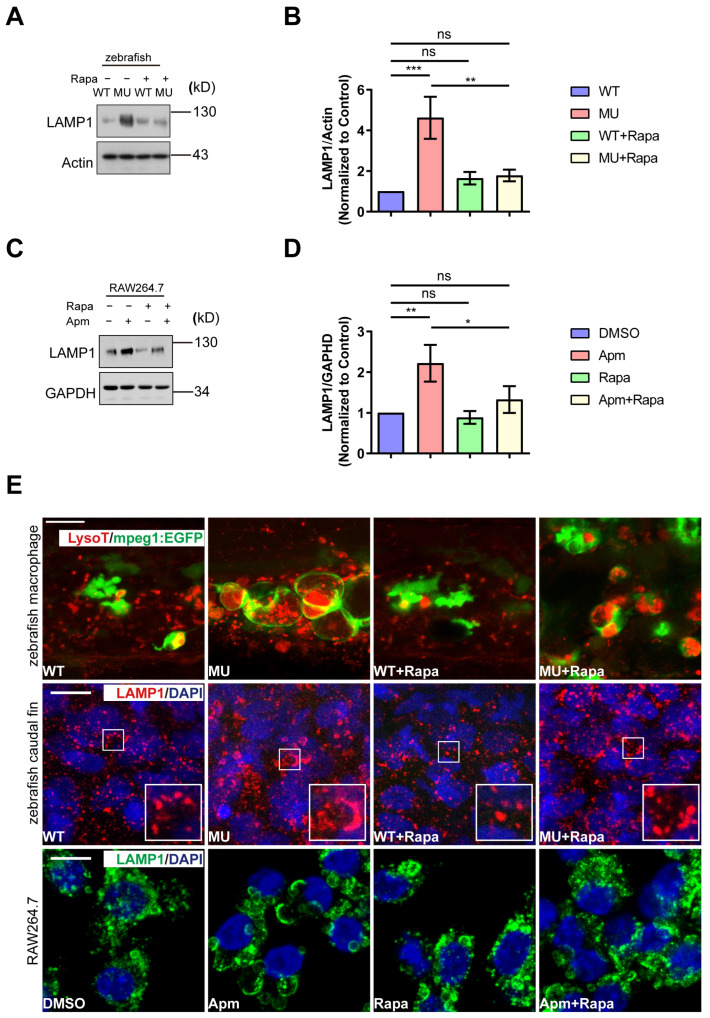
Rapamycin treatment ameliorates the lysosomal defects induced by PIKfyve deficiency in vivo and in vitro. (**A**–**D**) Western blot analysis showing the rapamycin treatment restored the elevated LAMP1 level induced by PIKfyve deficiency in zebrafish (**A**,**B**) and RAW264.7 cells (**C**,**D**). (**B**) and (**D**) are densitometric analysis of phosphorylation status of 4E-BP1 and S6K in (**A**) and (**C**), respectively (ns, non-significant; * *p* < 0.05, ** *p* < 0.01, *** *p* < 0.001; one-way ANOVA). (**E**) Lysotracker and LAMP1 staining in zebrafish and RAW264.7 cells. The Lysotracker/LAMP1-positive vacuoles were significantly enlarged in PIKfyve mutant cells and in Apm-treated cells, which were ameliorated by rapamycin treatment.

**Figure 6 cells-13-00953-f006:**
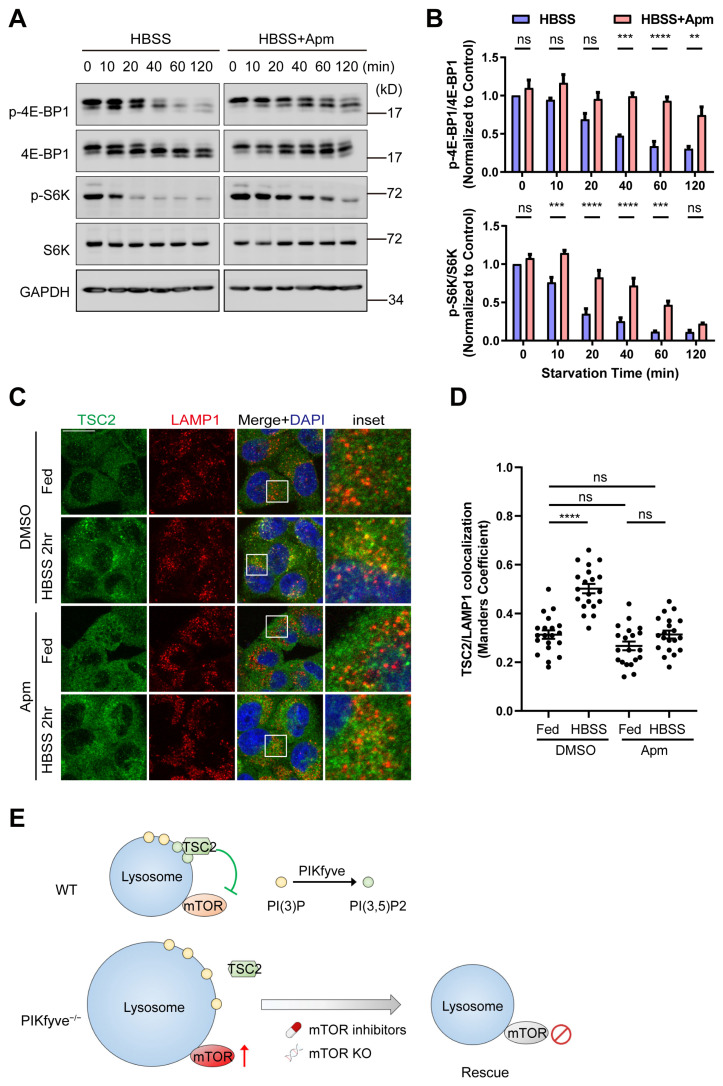
PIKfyve is required for starvation-induced mTOR shutdown. (**A**) HeLa cells were pretreated with DMSO vehicle control or 200 nM Apm for 3 h, followed by starvation in HBSS in the absence or in the presence of 200 nM Apm (HBSS + Apm) for the indicated time. Lysates were probed with the indicated antibodies. (**B**) Densitometric analysis of phosphorylation status of 4E-BP1 and S6K in (**A**). Data are represented as mean ± SD of 3 independent experiments (ns, non-significant; ** *p* < 0.01, *** *p* < 0.001, **** *p* < 0.0001; two-way ANOVA). (**C**) HeLa cells were treated as in (**A**), fixed at the selected time point 0 (Fed) and 120 min (HBSS 2 h). Cells were stained with the indicated antibodies and visualized by confocal microscopy. Scale bar, 20 µm. (**D**) Quantification of TSC2/LAMP1 colocalization from (**C**) (**** *p* < 0.0001, one-way ANOVA). (**E**) A working model for PIKfyve-dependent regulation of mTOR activity. The red arrow pointing upward indicates hyperactivation of mTOR activity. The prohibition sign indicates mTOR inhibition.

## Data Availability

Data will be made available on request.

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
