# Peer review of "Inhibition of PIKfyve Leads to Lysosomal Disorders via Dysregulation of mTOR Signaling"

_cells, 2024, doi:10.3390/cells13110953_

Round 1

Reviewer 1 Report

Comments and Suggestions for Authors

In this manuscript the authors propose to study the role PIKfyve and its effect on mTOR in related lysosomal phenotype.  The authors use genetically modified zebra danios to observe PIKfyve related changes in macrophage size and lysosome structure.  They further use transgenic danios which express eGFP in either the macrophage or the neutrophil.  While the studies are informative several concerns reduce enthusiasm for this manuscript.  Primarily the lack of the appropriate controls throughout the manuscript.  Further, the macrophages being studies should be assayed ex vivo and no in the context of the fish embryo.

Major comments:

1.     Figure1.  The description of the over phenotype demonstrated by the mutant/knockout fish is overstated.  The alterations in pigmentation are not consistent as they are apparent in MU1 but not MU2.  Further, the appearance of the structures in the CHT may or may not be vacuoles.  Additional studies should be done to confirm the identity of these structures. 

2.     CrisprCas9 is being used to generate proposed knockouts. Some assay to measure changes in protein should be performed.  Further, a direct readout should be performed showing that the activity of these mutants has truly been impacted, for instance the PIP2 phosphorylation.

3.     In B, fold change is being shown; however, the “WT” is apparently not the control.  What is the control being used to determine “1”.  Further, the knockouts demonstrate around 60% knockout.  In most studies this threshold may not be enough to measure physiological effects. 

4.     How are the cell diameters being measured.  If they are measured in the tissue the relative shape could be affected by extravasation or other such cellular movement processes.  Direct measurement assays should be performed on isolated purified populations to determine cell diameter.

5.     Figure 3, While the data show that the therapeutic targeting of mTOR reduced cell size, no information is provided showing whether this is a general effect of this strategy versus being specific for the mutant.  The effects of inhibitors on WT macrophage diameter should be included.

6.     Figure 3B.  Are the present changes in survival significant?

7.     Figure 5, the legend is not correct for the provided Figure.

8.     Dose escalations should be performed for the inhibitors being used to ensure that the doses used are not toxic and/or inducing artifacts that may not be physiologically relevant.

Minor Comments

1.     There is no data in Figure 3 showing the effectiveness of mTOR knockdown on mTOR expression.  Again, does mTOR KD has an effect on WT macrophages?

2.     Figure 4E, there is no representation of what happens to WT macrophages in vivo when treated with Rapamycin.  This should be included as a control for rapamycin as it has been shown to have pronounced off target effects. 

3.     There is no discussion of the differences of the targeting strategies as Torin-1 and mTOR KD would block both C1 and C2 whereas rapamycin is specific for C1.  

Comments on the Quality of English Language

The English is appropriate.

Author Response

Reviewer 1:

In this manuscript the authors propose to study the role PIKfyve and its effect on mTOR in related lysosomal phenotype. The authors use genetically modified zebra danios to observe PIKfyve related changes in macrophage size and lysosome structure. They further use transgenic danios which express eGFP in either the macrophage or the neutrophil. While the studies are informative several concerns reduce enthusiasm for this manuscript. Primarily the lack of the appropriate controls throughout the manuscript. Further, the macrophages being studies should be assayed ex vivo and no in the context of the fish embryo.

Major comments:

  1. Figure1. The description of the over phenotype demonstrated by the mutant/knockout fish is overstated. The alterations in pigmentation are not consistent as they are apparent in MU1 but not MU2. Further, the appearance of the structures in the CHT may or may not be vacuoles. Additional studies should be done to confirm the identity of these structures.

Our response:

We acknowledge that the alterations in pigmentation are not consistent across all mutants, as evident in the comparison between MU1 and MU2. In the revised manuscript, the overstated description of the pigmentation alterations was deleted.

We appreciate the reviewer's suggestion to conduct additional studies to confirm the identity of the vacuole-like structures observed in the CHT to avoid potential misinterpretation. In the revised manuscript, “giant vacuoles” was replaced with “giant vacuole-like structures” for a more accurate description. We plan to conduct further investigations, including histological analyses and marker antibody/dye staining, to characterize the vacuole-like structures observed in the mutant CHT in detail.

  1. CrisprCas9 is being used to generate proposed knockouts. Some assay to measure changes in protein should be performed. Further, a direct readout should be performed showing that the activity of these mutants has truly been impacted, for instance the PIP2 phosphorylation.

Our response:

We thank the reviewer to point out this important issue. We performed some of the suggested experiments but unfortunately failed to get convincing data. The following was discussed in the revised manuscript:

(1) Western blot to measure the changes in protein.

To measure the changes of PIKfyve and mTOR protein in the mutants, Western blot was applied in lysates from WT and MU zebrafish embryos. Compared to mammalian models, zebrafish research currently suffers from a lack of specific antibodies. We have tested three commercially available anti-human PIKfyve antibodies (Invitrogen PA5-67981, Abnova H00200576-M01, SantaCruz sc-100408) and one mTOR antibody (CST, #2983), but none of them exhibit cross-reactivity with zebrafish samples. So we could not experimentally verify the changes of target protein in PIKfyve or mTOR mutants.  

(2) Direct readout of PIKfyve activity.

We plan to perform in vitro kinase assays by measuring PI(3,5)P2 levels using zebrafish lysates [1]. In addition, referring to the previously published PI(3)P reporter system in our lab [2], we designed a fluorescent reporter system for real-time monitoring of PI(3,5)P2 in zebrafish embryos. A transgenic zebrafish line expressing the PI(3,5)P2 biosensor (GFP-2xMLN1) [3] driven by the ubiquitin promoter is being constructed. These assays will provide a direct readout of PIKfyve enzymatic activity in WT and MU zebrafish larvae, allowing us to validate the functional impact of PIKfyve knockout.

  1. In B, fold change is being shown; however, the “WT” is apparently not the control. What is the control being used to determine “1”. Further, the knockouts demonstrate around 60% knockout. In most studies this threshold may not be enough to measure physiological effects.

Our response:

Thank you for your insightful comments and observations.

(1) Control for fold change.

We apologize for the lack of clarity regarding the control used to determine the fold change in Figure 1B. WT siblings and mutants were collected from the same clutch and separated after genotyping [4]. WT siblings were considered to be an appropriate control. To be more specific, F2 larvae from heterozygous F1 incrosses were collected, fin clipped, and genotyped. The rest of the body was kept in TRIzol Reagent in individual PCR tubes. Once the genotype was determined, embryos were grouped into WT siblings, heterozygous and homozygous mutants yielding 3 groups from 3 different clutches (3 biological replicates). Note that all 3 genotypes were from the same clutch, so all of them were siblings for each experiment. In the revised manuscript, detailed methods of sample grouping and preparation were added in section 2.7 “Real-time quantitative PCR analysis”.

(2) Threshold for knockout efficiency:

We appreciate your concern regarding the knockout efficiency threshold and its implications for measuring physiological effects. Firstly, in this study the aim of gene editing is to introduce frameshift mutations in the key kinase domain (PIPKc domain, Supplementary Figure S1). In MU embryos, even if the frameshifted transcripts are present, they cannot be translated into a functional protein. On the other hand, RT-qPCR measures the efficiency of nonsense-mediated mRNA decay (NMD), which is highly variable among different cell lines and mRNAs [5,6]. NMD is an evolutionarily conserved mRNA degradation pathway that eliminates mRNAs with a premature termination codon (PTC) arising from mutations, alternative splicing, or other events in cells[7]. High throughput systematic analysis of half-lives of the aberrant mRNAs containing PTCs demonstrate that a small percent escape surveillance and do not degrade [8]. In this study, the remaining 40% of PIKfyve transcripts with PTCs may survive due to evasion of NMD mechanism, whereas they were unable to translate into a functional protein.

  1. How are the cell diameters being measured? If they are measured in the tissue the relative shape could be affected by extravasation or other such cellular movement processes. Direct measurement assays should be performed on isolated purified populations to determine cell diameter.

Our response:

We appreciate your attention to the methodology and potential confounding factors in cell diameter measurements. We acknowledge the importance of accurate cell diameter measurements and the potential influence of tissue environment on cellular morphology. In our study, cell diameters were measured using ImageJ software in fluorescence images taken by a confocal laser microscope [9]. In response to your suggestion, we plan to perform additional experiments using isolated and purified cell populations by FACS to directly measure cell diameter. This approach will allow us to mitigate any potential effects of tissue environment on cell shape and provide more precise measurements.

  1. Figure 3, while the data show that the therapeutic targeting of mTOR reduced cell size, no information is provided showing whether this is a general effect of this strategy versus being specific for the mutant. The effects of inhibitors on WT should be included.

Our response:

Thank you for your valuable suggestion! We performed the suggested experiments in WT siblings and found that macrophage diameter in WT siblings was not affected by pharmacological or genetic inhibition of mTOR (revised Supplementary Figure S2). 

  1. Figure 3B. Are the present changes in survival significant?

Our response:

We performed statistical analysis on the survival curves and the results showed significant differences (p<0.05). The figure and legend of Figure 3B have been revised accordingly.

  1. Figure 5, the legend is not correct for the provided Figure.

Our response:

We apologize for the mistake and thanks for your suggestion. The legend has been revised.

  1. Dose escalations should be performed for the inhibitors being used to ensure that the doses used are not toxic and/or inducing artifacts that may not be physiologically relevant.

Our response:

Thanks for your suggestion. We set different drug concentrations and different incubation time windows for each inhibitor being used. To avoid toxic artifacts, we chose the combination of the lowest drug concentration and the shortest time window within the linear range that enables credible quantification.

Minor Comments

  1. There is no data in Figure 3 showing the effectiveness of mTOR knockdown on mTOR expression. Again, does mTOR KD has an effect on WT macrophages?

Our response:

Thanks for your suggestions!

(1) Effectiveness of mTOR knockdown on mTOR expression.

mTOR gRNA was synthesized according to a published report [10]. We provided evidence for efficient gene editing in Supplementary Figure S3. Due to the lack of working antibodies against zebrafish mTOR, we tried but failed to obtain valid data from Western blot.

(2) Effects of mTOR Knockout on WT macrophages.

As revised in Supplementary Figure S2, macrophage diameter in WT siblings was not affected by genetic mTOR knockout. 

  1. Figure 4E, there is no representation of what happens to WT macrophages in vivo when treated with Rapamycin. This should be included as a control for rapamycin as it has been shown to have pronounced off target effects.

Our response:

Thanks for your suggestion! Representative image of WT macrophages treated with rapamycin has been added in Figure 4E.

  1. There is no discussion of the differences of the targeting strategies as Torin-1 and mTOR KD would block both C1 and C2 whereas rapamycin is specific for C1.

Our response:

Thanks for your valuable suggestions. Our study focuses on elucidating the role of PIKfyve and its effect on mTOR signaling pathways in macrophages. One important aspect to consider is the diverse targeting strategies employed to inhibit mTOR activity, each with distinct mechanisms of action. Specifically, Torin-1 is a potent and selective ATP-competitive inhibitor of mTOR kinase activity, effectively blocking the function of both C1 and C2. In contrast, rapamycin primarily targets C1 and has limited efficacy in inhibiting C2. Moreover, the differential effects of these inhibitors on mTORC1 and mTORC2 could impact cellular responses in a context-dependent manner. For instance, the specific inhibition of mTORC1 by rapamycin may lead to distinct physiological outcomes compared to the broader inhibition of both complexes by Torin-1 or mTOR KD. Understanding these differences is crucial for interpreting experimental results and designing targeted interventions to modulate mTOR signaling effectively.

In our experiments, two commonly used mTOR inhibitors as well as genetic mTOR knockout obtained similar rescue results, which increased the reliability of the conclusions. Further investigation into the specific effects of each inhibitor on mTOR signaling pathways will provide valuable insights into their therapeutic potential and biological consequences.

References:

  1. Wible, D.J.; Parikh, Z.; Cho, E.J.; Chen, M.-D.; Jeter, C.R.; Mukhopadhyay, S.; Dalby, K.N.; Varadarajan, S.; Bratton, S.B. Unexpected Inhibition of the Lipid Kinase PIKfyve Reveals an Epistatic Role for P38 MAPKs in Endolysosomal Fission and Volume Control. Cell Death Dis. 2024, 15, 80, doi:10.1038/s41419-024-06423-0.
  2. Zhao, S.; Xia, J.; Wu, X.; Zhang, L.; Wang, P.; Wang, H.; Li, H.; Wang, X.; Chen, Y.; Agnetti, J.; et al. Deficiency in Class III PI3-Kinase Confers Postnatal Lethality with IBD-like Features in Zebrafish. Nat. Commun. 2018, 9, 2639, doi:10.1038/s41467-018-05105-8.
  3. Li, X.; Wang, X.; Zhang, X.; Zhao, M.; Tsang, W.L.; Zhang, Y.; Yau, R.G.W.; Weisman, L.S.; Xu, H. Genetically Encoded Fluorescent Probe to Visualize Intracellular Phosphatidylinositol 3,5-Bisphosphate Localization and Dynamics. Proc. Natl. Acad. Sci. 2013, 110, 21165–21170, doi:10.1073/pnas.1311864110.
  4. Dupret, B.; Völkel, P.; Follet, P.; Le Bourhis, X.; Angrand, P.-O. Combining Genotypic and Phenotypic Analyses on Single Mutant Zebrafish Larvae. MethodsX 2018, 5, 244–256, doi:10.1016/j.mex.2018.03.002.
  5. Sato, H.; Singer, R.H. Cellular Variability of Nonsense-Mediated MRNA Decay. Nat. Commun. 2021, 12, 7203, doi:10.1038/s41467-021-27423-0.
  6. Lindeboom, R.G.H.; Vermeulen, M.; Lehner, B.; Supek, F. The Impact of Nonsense-Mediated MRNA Decay on Genetic Disease, Gene Editing and Cancer Immunotherapy. Nat. Genet. 2019, 51, 1645–1651, doi:10.1038/s41588-019-0517-5.
  7. El-Brolosy, M.A.; Kontarakis, Z.; Rossi, A.; Kuenne, C.; Günther, S.; Fukuda, N.; Kikhi, K.; Boezio, G.L.M.; Takacs, C.M.; Lai, S.-L.; et al. Genetic Compensation Triggered by Mutant MRNA Degradation. Nature 2019, 568, 193–197, doi:10.1038/s41586-019-1064-z.
  8. Lejeune, F. Nonsense-Mediated MRNA Decay, a Finely Regulated Mechanism. Biomedicines 2022, 10, 141, doi:10.3390/biomedicines10010141.
  9. Berg, R.D.; Levitte, S.; O’Sullivan, M.P.; O’Leary, S.M.; Cambier, C.J.; Cameron, J.; Takaki, K.K.; Moens, C.B.; Tobin, D.M.; Keane, J.; et al. Lysosomal Disorders Drive Susceptibility to Tuberculosis by Compromising Macrophage Migration. Cell 2016, 165, 139–152, doi:10.1016/j.cell.2016.02.034.
  10. Bu, H.; Ding, Y.; Li, J.; Zhu, P.; Shih, Y.-H.; Wang, M.; Zhang, Y.; Lin, X.; Xu, X. Inhibition of MTOR or MAPK Ameliorates Vmhcl/Myh7 Cardiomyopathy in Zebrafish. JCI Insight 2021, 6, doi:10.1172/jci.insight.154215.

Reviewer 2 Report

Comments and Suggestions for Authors

The manuscript shows that PIKfyve-deficient zebrafish embryos exhibit enlarged macrophages with giant vacuoles, suggesting lysosomal stress; inhibition of mTOR partially mitigates the stress and extend the lifespan of mutant larvae. They also show that PIKfyve activity is essential for mTOR inactivation during early zebrafish development and in serum-starved cells. The research is interesting. Before the manuscript is suggested for publication in the journal, some concerns need to be addressed.

Major concerns:

1)    Fig. 1B shows that PIKfyve mRNA expression levels were downregulated by approximately 60% using CRISPR/Cas9 technology. It is suggested to explain why PIKfyve was not completely knocked out.

2)    Fig. 1C-F show that “the macrophages were enlarged with giant vacuoles while the neutrophils appeared normal”. Some discussions on this are suggested.

3)    For Fig. 3, please include a Western blot showing that mTOR is knocked out in the mutants. Also, it is unknown whether a sgRNA control was used, related to mTOR-KO.

4)    The authors show that the enlarged macrophages with giant vacuoles in PIKfyve mutants were identified as fused lysosomes (Fig. 2A-D); hyperactive mTORC1 signaling in PIKfyve mutant zebrafish (Fig. 4); inhibition of mTOR mitigates the lysosome stress induced by PIKfyve inhibition in zebrafish macrophages in vivo and RAW264.7 cells in vitro (Fig. 3 and Fig. 5). mTOR is a negative regulator of autophagy. It is good to determine if defected autophagy occurs in the macrophages of PIKfyve mutants, leading to the lysosome stress.

5)    It has been shown that VPS34 forms a protein complex with PIKfyve and TSC1, which disrupts the TSC1/TSC2 complex, resulting in ubiquitination and degradation of TSC2 and consequent activation of the Rheb-mTORC1 axis [PMID: 27409169]. Fig.6A and B show that inhibition of PIKfyve with Apilimod increases the phosphorylation of S6K and 4E-BP1 in serum-starved HeLa cells. This may involve TSC2 (Fig. 6C-E). It would be better to have a further study to determine if inhibition of PIKfyve results in activation of mTORC1 by the mechanism abovementioned.

6)    Studies have shown that inhibition of PIKfyve induces lysosome-associated cytoplasmic vacuolation in hepatocellular carcinoma cells (PMID: 33031902). Also, glucose starvation can induce autophagy by ULK1-mediated activation of PIKfyve (PMID: 34107300), and trehalose can activate PIKFYVE leading to TFEB nuclear translocation leading to autophagy (PMID: 32079455). So, more discussions on the role of PIKFYVE in autophagy are suggested.

Minor concerns:

1)    Please define all abbreviations at the first appearance in the main text, such as hpf, PBS, DMEM, DMSO, RT, …, although some of these abbreviations have been provided at the very beginning.

2)    Please provide more detailed information about all reagents and instruments, including vendor/company, city, province/state, country.

3)    Change “… leading mTOR activation…” (Line 77) to “… leading to mTOR activation…).

4)    Is “Figure 1. heterozygous fish were…” (Lines 215-224) also a figure legend? If not, please state it more clearly.

5)    Please describe clearly how to evaluate cell death/degeneration in the brain (Figure 1A, asterisk), pharynx, esophagus and intestinal tract (related to Lines 219-220).

6)    It is good to label some Neutral Red (NR)-positive microglial cells using arrows in Fig. 1G.

7)    Change “Figure 3E, F” (Line 269) to “Figure 2E, F).

Comments on the Quality of English Language

See the above comments.

Author Response

Reviewer 2:

The manuscript shows that PIKfyve-deficient zebrafish embryos exhibit enlarged macrophages with giant vacuoles, suggesting lysosomal stress; inhibition of mTOR partially mitigates the stress and extend the lifespan of mutant larvae. They also show that PIKfyve activity is essential for mTOR inactivation during early zebrafish development and in serum-starved cells. The research is interesting. Before the manuscript is suggested for publication in the journal, some concerns need to be addressed.

Major concerns:

1) Fig. 1B shows that PIKfyve mRNA expression levels were downregulated by approximately 60% using CRISPR/Cas9 technology. It is suggested to explain why PIKfyve was not completely knocked out.

Our response:

We appreciate your concern regarding the knockout efficiency threshold. We have included a discussion in the revised manuscript to address this concern.

While CRISPR/Cas9 technology is powerful for gene editing, achieving complete knockout can be challenging due to various factors such as off-target effects, incomplete editing efficiency, and genetic compensation mechanisms. Firstly, in this study the aim of gene editing is to introduce frame-shift mutations in the key kinase domain (PIPKc domain, Supplementary Figure S1). In MU embryos, even if the frame-shifted transcripts are present, they cannot be translated into a functional protein. On the other hand, RT-qPCR measures the efficiency of nonsense-mediated mRNA decay (NMD), which is highly variable among different cell lines and mRNAs [1]. NMD is an evolutionarily conserved mRNA degradation pathway that eliminates mRNAs with a premature termination codon (PTC) arising from mutations, alternative splicing, or other events in cells. High throughput systematic analysis of half-lives of the aberrant mRNAs containing PTCs demonstrate that a small percent escape surveillance and do not degrade [2]. In this study, the remaining 40% of PIKfyve transcripts with PTCs may survive due to evasion of NMD mechanism, whereas they were unable to translate into a functional protein.

2) Fig. 1C-F show that “the macrophages were enlarged with giant vacuoles while the neutrophils appeared normal”. Some discussions on this are suggested.

Our response:

Thanks for highlighting this observation. The differential response of macrophages and neutrophils to PIKfyve deficiency suggests that PIKfyve and its lipid products may have different effects on distinct immune cell populations, potentially reflecting differences in their lysosomal physiology, metabolic demands, or signaling pathways.

3) For Fig. 3, please include a Western blot showing that mTOR is knocked out in the mutants. Also, it is unknown whether a sgRNA control was used, related to mTOR-KO.

Our response:

Thank you for your valuable suggestions.

(1) Western Blot for mTOR Knockout Confirmation: We acknowledge the importance of validating mTOR knockout in the mutants through Western blot analysis. Unfortunately, due to the lack of working antibodies against zebrafish mTOR, we tried but failed to obtain valid data from Western blot. In this study, mTOR gRNA was synthesized according to a published report [3]. We have provided evidence for efficient gene editing in Supplementary Figure S3.

(2) Inclusion of sgRNA Control: We apologize for the lack of clarity regarding the gRNA control. In this study, larvae injected with Cas9 protein alone (with no gRNA) were used as control related to mTOR KO. According to your suggestion, we will design a non-targeting gRNA control to ensure specificity of gene editing.

4) The authors show that the enlarged macrophages with giant vacuoles in PIKfyve mutants were identified as fused lysosomes (Fig. 2A-D); hyperactive mTORC1 signaling in PIKfyve mutant zebrafish (Fig. 4); inhibition of mTOR mitigates the lysosome stress induced by PIKfyve inhibition in zebrafish macrophages in vivo and RAW264.7 cells in vitro (Fig. 3 and Fig. 5). mTOR is a negative regulator of autophagy. It is good to determine if defected autophagy occurs in the macrophages of PIKfyve mutants, leading to the lysosome stress.

Our response:

Thank you for your insightful suggestion regarding the potential involvement of defective autophagy in the macrophages of PIKfyve mutants, contributing to lysosome stress. We will further investigate this issue. Here's our experiment plan:

(1) Evaluation of Autophagy Markers: We plan to assess the status of autophagy in macrophages of PIKfyve MU compared to WT siblings. This will involve examining key markers of autophagy, such as LC3 lipidation and p62 degradation, through Western blot analysis or immunofluorescence staining. By quantifying the levels of these markers, we aim to determine if there are alterations in autophagic activity in PIKfyve-deficient macrophages.

(2) Functional Assays for Autophagy Flux: To further elucidate the dynamics of autophagy in PIKfyve mutants, we will conduct functional assays to measure autophagy flux. This may involve using fluorescent reporters or flux inhibitors to monitor the degradation and turnover of autophagic substrates.

(3) Correlation with Lysosome Stress: We will analyze the relationship between defective autophagy and lysosome stress in PIKfyve mutants. By correlating changes in autophagy markers with the presence of enlarged lysosomes and vacuoles, we can better understand the mechanistic link between these processes and their contribution to macrophage dysfunction in PIKfyve deficiency.

We plan to carry out these additional experiments in the near future, which will provide a more comprehensive understanding of the role of autophagy in mediating lysosome stress in PIKfyve mutant macrophages.

5) It has been shown that VPS34 forms a protein complex with PIKfyve and TSC1, which disrupts the TSC1/TSC2 complex, resulting in ubiquitination and degradation of TSC2 and consequent activation of the Rheb-mTORC1 axis [PMID: 27409169]. Fig.6A and B show that inhibition of PIKfyve with Apilimod increases the phosphorylation of S6K and 4E-BP1 in serum-starved HeLa cells. This may involve TSC2 (Fig. 6C-E). It would be better to have a further study to determine if inhibition of PIKfyve results in activation of mTORC1 by the mechanism abovementioned.

Our response:

We appreciate your valuable suggestions highlighting the potential mechanism by which inhibition of PIKfyve could lead to activation of the mTORC1 pathway via disruption of the TSC1/TSC2 complex. Here's how we plan to address your suggestions:

(1) Evaluation of TSC1/TSC2 Complex Disruption: We will test the effect of PIKfyve inhibition on the integrity of the TSC1/TSC2 complex. This may involve co-immunoprecipitation assays to examine the interaction between TSC1 and TSC2 in the presence or absence of PIKfyve inhibitors. By quantifying changes in the stability or association of these proteins, we aim to determine if PIKfyve inhibition disrupts the TSC1/TSC2 complex.

(2) Analysis of Rheb-mTORC1 axis: Following inhibition of PIKfyve, we will evaluate the activation status of the Rheb-mTORC1 pathway. Additionally, we will investigate whether these changes in mTORC1 signaling are dependent on TSC2 by comparing the response in WT and TSC2-deficient cells.

By conducting these experiments, we aim to provide further insights into the molecular mechanisms underlying the activation of mTORC1 following inhibition of PIKfyve.

6) Studies have shown that inhibition of PIKfyve induces lysosome-associated cytoplasmic vacuolation in hepatocellular carcinoma cells (PMID: 33031902). Also, glucose starvation can induce autophagy by ULK1-mediated activation of PIKfyve (PMID: 34107300), and trehalose can activate PIKFYVE leading to TFEB nuclear translocation leading to autophagy (PMID: 32079455). So, more discussions on the role of PIKFYVE in autophagy are suggested.

Our response:

Thank you for highlighting the role of PIKfyve in autophagy regulation and its implications in various cellular contexts. We added the discussion in the revised manuscript as followed:

PIKfyve plays a crucial role in regulating intracellular membrane trafficking and organelle homeostasis. In recent years, accumulating evidence has highlighted the significance of PIKfyve in autophagy regulation, a highly conserved cellular process involved in the degradation and recycling of cellular components. Inhibition of PIKfyve induces lysosome-associated cytoplasmic vacuolation in hepatocellular carcinoma cells. Glucose starvation can induce autophagy by ULK1-mediated activation of PIKfyve [4], and trehalose can activate PIKfyve leading to TFEB nuclear translocation and autophagy induction [5]. PIKfyve has been implicated in the regulation of TFEB, a master regulator of lysosomal biogenesis and autophagy. Studies have shown that PIKfyve inhibition leads to mTORC1-dependent phosphorylation and cytoplasmic retention of TFEB, thereby inhibiting its transcriptional activity and impairing lysosome biogenesis. Conversely, activation of PIKfyve promotes TFEB nuclear translocation and enhances autophagic activity. These findings underscore the multifaceted role of PIKfyve in coordinating cellular responses to nutrient availability and stress, with implications for autophagy induction and lysosome function. Given the emerging role of autophagy dysregulation in various diseases, including cancer and neurodegenerative disorders, understanding the interplay between PIKfyve and autophagy pathways may offer novel opportunities for therapeutic intervention.  

Minor concerns:

1) Please define all abbreviations at the first appearance in the main text, such as hpf, PBS, DMEM, DMSO, RT,  …, although some of these abbreviations have been provided at the very beginning.

Our response:

Thanks for your suggestion! We agree with the reviewer and have now defined all abbreviations at the first appearance in the main text.

2) Please provide more detailed information about all reagents and instruments, including vendor/company, city, province/state, country.

Our response:

Thanks for your suggestion! We have added more detailed information about all reagents and instruments.

3) Change “… leading mTOR activation…” (Line 77) to “… leading to mTOR activation…).

Our response:

We sincerely thank the reviewer for careful reading. We have revised this sentence as suggested.

4) Is “Figure 1. heterozygous fish were…” (Lines 215-224) also a figure legend? If not, please state it more clearly.

Our response:

We apologize for this mistake and have corrected it in the revised manuscript.

5) Please describe clearly how to evaluate cell death/degeneration in the brain (Figure 1A, asterisk), pharynx, esophagus and intestinal tract (related to Lines 219-220).

Our response:

Cell death was visualized by reduction of transparency in these tissues under a stereoscope [6].

6) It is good to label some Neutral Red (NR)-positive microglial cells using arrows in Fig. 1G.

Our response:

Thanks for your suggestion! Arrows have been added to label some NR-positive microglial cells as suggested.

7) Change “Figure 3E, F” (Line 269) to “Figure 2E, F).

Our response:

We apologize for this oversight and have corrected it. Additionally, we have carefully revised the manuscript to eliminate any further errors. Thank you for careful reading.

  1. Sato, H.; Singer, R.H. Cellular Variability of Nonsense-Mediated MRNA Decay. Nat. Commun. 2021, 12, 7203, doi:10.1038/s41467-021-27423-0.
  2. Lejeune, F. Nonsense-Mediated MRNA Decay, a Finely Regulated Mechanism. Biomedicines 2022, 10, 141, doi:10.3390/biomedicines10010141.
  3. Bu, H.; Ding, Y.; Li, J.; Zhu, P.; Shih, Y.-H.; Wang, M.; Zhang, Y.; Lin, X.; Xu, X. Inhibition of MTOR or MAPK Ameliorates Vmhcl/Myh7 Cardiomyopathy in Zebrafish. JCI Insight 2021, 6, doi:10.1172/jci.insight.154215.
  4. Karabiyik, C.; Vicinanza, M.; Son, S.M.; Rubinsztein, D.C. Glucose Starvation Induces Autophagy via ULK1-Mediated Activation of PIKfyve in an AMPK-Dependent Manner. Dev. Cell 2021, 56, 1961-1975.e5, doi:10.1016/j.devcel.2021.05.010.
  5. Sharma, V.; Makhdoomi, M.; Singh, L.; Kumar, P.; Khan, N.; Singh, S.; Verma, H.N.; Luthra, K.; Sarkar, S.; Kumar, D. Trehalose Limits Opportunistic Mycobacterial Survival during HIV Co-Infection by Reversing HIV-Mediated Autophagy Block. Autophagy 2021, 17, 476–495, doi:10.1080/15548627.2020.1725374.
  6. Nam, H.-S.; Hwang, K.-S.; Jeong, Y.-M.; Ryu, J.-I.; Choi, T.-Y.; Bae, M.-A.; Son, W.-C.; You, K.-H.; Son, H.-Y.; Kim, C.-H. Expression of MiRNA-122 Induced by Liver Toxicants in Zebrafish. Biomed Res. Int. 2016, 2016, 1–7, doi:10.1155/2016/1473578.

Round 2

Reviewer 2 Report

Comments and Suggestions for Authors

The manuscript has been well revised.